# MULTIMODAL GENERATIVE AI FOR STORY POINT ESTIMATION

## ABSTRACT

This research explores the application of Multimodal Generative AI to enhance story point estimation in Agile software development. By integrating text, image, and categorical data using advanced models like BERT, CNN, and XGBoost, our approach surpasses the limitations of traditional single-modal estimation methods. The results demonstrate good accuracy for simpler story points, while also highlighting challenges in more complex categories due to data imbalance. This study further explores the impact of categorical data, particularly severity, on the estimation process, emphasizing its influence on model performance. Our findings emphasize the transformative potential of multimodal data integration in refining AI-driven project management, paving the way for more precise, adaptable, and domain-specific AI capabilities. Additionally, this work outlines future directions for addressing data variability and enhancing the robustness of AI in Agile methodologies.

## 1   INTRODUCTION

Story points (SP) are a key metric in Agile methodologies used to estimate the size, complexity, and effort for each user story, which is a brief description of a software feature from the end user's perspective, outlining their needs and reasons. They are also employed to estimate the remaining useful life of software products (Islam & Sandborn, 2021, 2023, 2024). Agile teams typically use subjective methods such as planning poker to estimate these points, but this process often exhibits inconsistency and variable accuracy (Jorgensen, 2001 & Usman et al., 2014). The inherent complexity of software development within Agile frameworks demands more precise and adaptable techniques for estimating story points (Menzies et al., 2006). Recent advancements in Generative AI, particularly multimodal models that integrate various data formats such as text, images, graphs, and categorical data, present a groundbreaking solution to these challenges (Devlin et al., 2019; He et al., 2016). Deep learning architectures in these models process and integrate multimodal inputs, enabling a more nuanced analysis of text-based data and resulting in predictions that are both more accurate and consistent (Radford et al., 2021). Multimodal Generative AI exploits the synergistic potential of diverse data types, uncovering complex relationships among textual descriptions, visual elements, historical data, and categorical features. This comprehensive approach not only improves the accuracy of story point estimation, aligning with Agile principles, but also enhances the responsiveness and adaptability of the development process (Vaswani et al., 2017). Integrating these models within software development workflows reduces human bias and shortens project timelines, leading to substantial cost savings by minimizing delays and avoiding unnecessary rework (Lin et al., 2014).

This paper proposes a novel framework that uses state-of-the-art multimodal machine learning techniques, including Ordinal Encoding, BERT (Bidirectional Encoder Representations from Transformers), CNN (Convolutional Neural Networks), XGBoost (Extreme Gradient Boosting), and other models, to refine the task of story point estimation. Through empirical analysis, we aim to show how multimodal Generative AI can significantly advance Agile software development by effectively addressing the complexities associated with story point estimation. Our findings support the adoption of these technologies to foster more reliable, consistent, and adaptable development practices, setting a new benchmark for future advancements in the field.

## 2 RELATED WORK

Researchers have extensively studied the field of story point estimation within Agile software development, with traditional approaches predominantly relying on expert judgment, historical data analysis, and machine learning techniques such as regression models and decision trees. While useful, these methods often struggle with inconsistencies and inaccuracies due to their reliance on single-modal data inputs, such as text descriptions of user stories (Friedman, 2001). Recent advances in machine learning, particularly with the advent of deep learning and natural language processing (NLP), have introduced more sophisticated approaches. However, even these advanced techniques face limitations in integrating the diverse data types often present in software development processes.

One significant development in machine learning has been the adoption of Generative AI models, particularly those based on transformer architectures, to enhance the accuracy of story point estimation. Models like BERT (Devlin et al., 2019) and GPT (Brown et al., 2020) have demonstrated promise in processing textual data and capturing the nuances of user stories with a level of detail previously unattainable. However, these models typically focus solely on textual analysis and do not fully exploit the potential of multimodal data integration, limiting their effectiveness in contexts where visual or categorical data are also relevant. Multimodal learning has emerged as a promising approach to overcome these limitations by integrating various data formats such as text, images, graphs, and categorical data. Research in this domain has shown that multimodal models can capture more complex relationships between different types of data, leading to improved performance in tasks like image captioning (Radford et al., 2021), sentiment analysis (Wang & Deng, 2018), and medical diagnosis (Wang et al., 2020). Despite these advancements, applying multimodal learning to story point estimation in Agile software development remains underexplored. Our work builds upon these foundations by introducing a Multimodal Generative AI approach that integrates not only textual but also visual and categorical data, thereby creating a more comprehensive and accurate estimation model. Unlike previous single-modal methodologies, our framework leverages the strengths of multimodal integration, offering a holistic perspective of user stories and their inherent complexities. This approach promises a significant improvement over traditional methods by providing a deeper understanding of the multifaceted aspects of story points.

Addressing a critical gap in existing research, our study specifically tailors multimodal learning to the unique challenges of Agile methodologies, which require rapid iteration and adaptability. This customization ensures that our model integrates seamlessly into Agile workflows, delivering real-time, adaptive story point estimates. By extending multimodal learning techniques to Agile story point estimation, our paper advances the state of the art, overcoming previous limitations and illuminating new ways to incorporate diverse data types for more accurate and efficient software development practices. Our research presents a novel framework for integrating multimodal data into Agile software development, paving the way for more reliable, consistent, and adaptable practices. This framework makes a significant contribution to the field, offering a robust solution to the longstanding challenges of story point estimation.

## 3 OUR APPROACHES

### 3.1 DATA COLLECTION

For this research, we engaged in a comprehensive data collection process from Bugzilla[1], an opensource bug tracking system, to estimate story points in Agile software development. We chose Bugzilla for its opensource nature, which provides access to a vast record of historical user stories focused exclusively on fixes, enhancements, and tasks related to Bugzilla itself. This includes release-wise data and associated image data, such as wireframes and screenshots of errors. Additionally, Bugzilla offers relevant historical comments from multiple users. This rich data set provides the diverse and detailed information necessary for our analysis, making Bugzilla an ideal choice for this project. The data we collected was diverse, encompassing textual descriptions of user stories, historical data on story points previously assigned to similar user stories, and various visual aids such as UI/UX mockups, system architecture diagrams, screenshots of errors, and other relevant images like UI screenshots and flowcharts (Table 1). We collected categorical data encompassing variables such

---

[1]https://www.bugzilla.org/releases/

Table 1: User Stories with Severity and Story Points

| USER STORY | SEVERITY | SP | SCREENSHOT |
|---|---|---|---|
| Bugzilla cannot connect to Oracle 11G RAC | 2 | 2 | |
| Typing something like "P1-5" in the quicksearch box... | 2 | 5 | |
| Users who had passwords less than 6 characters long couldn't log in. | 1 | 2 | |
| A regression in Bugzilla 4.4.3 due to CVE-2014-1517... | 1 | 2 | |
| Update MySQL v5.5.5-10.3.7-MariaDB1:10.3.7+maria jessi | 1 | 3 | |
| Remove product and component from UNSUPPORTED FIELDS. | 1 | 2 | |

as severity levels (e.g., high, medium, low). Our proposed model classifies story points (SP) using the Fibonacci sequence, a widely adopted system known for its scalability and intuitive handling of task complexity and size in project management and software development. In this research, we used the industry-standard sequences of 1, 2, 3, 5, and 8, but additional sequences can be seamlessly integrated if needed. We also organized the collection of historical story point data for individual user stories as part of our comprehensive data gathering process.

We meticulously sourced the text data from Bugzilla repositories, involving the extraction and cleaning of raw textual descriptions of bugs and feature requests. Historical story points data provided insights into the assessment trends and valuation of similar past stories. We curated the image data from associated repositories to ensure a thorough compilation of visuals that contextualize the user stories, including system architecture, wireframes, UI/UX design wireframes, screenshots, and others. For the categorical data, we included attributes like severity levels to facilitate feature engineering and enhance the model's accuracy. To manage and streamline the workflow, we consolidated all collected data—text, graphs, images, and categorical inputs—into a unified dataset. Additionally, we utilized Pinecone, a vector database, to store and process the embedded data, ensuring organized storage and efficient handling of complex queries for subsequent analysis and modeling stages.

## 3.2 DATA PRE-PROCESSING AND FEATURE ENGINEERING

We meticulously pre-processed the raw data for this project to prepare it for use in machine learning models. We refined the text data by removing extraneous details, normalizing the language, and tokenizing the content, while pre-processing the image data involved resizing, normalization, and feature extraction to ensure effective representation of the visual and textual content in the form of embeddings. Our entire corpus consists of 113 observations. For feature extraction and embedding, we utilized BERT (Bidirectional Encoder Representations from Transformers) for text data and CNN (Convolutional Neural Networks) for image data. We chose BERT for its ability to understand the context within user stories, making it ideal for tasks requiring deep semantic comprehension, such as classification or sentiment analysis (Table 2). We selected CNNs for their exceptional ability to process and analyze visual data. Additionally, we applied ordinal encoding to categorical data such as severity and story points, leveraging the inherent order within these categories to enhance model interpretability. We used Fibonacci sequencing to estimate story points. Ordinal encoding is particularly valuable for encoding categorical features that follow a natural sequence or hierarchy, ensuring that the encoded data accurately reflects the structured relationships inherent in the project's categories. We integrated these processed features into a multimodal dataset ready for machine learning in the final step. This fusion combined cleaned text, image features, and encoded categorical data into a unified format. To facilitate effective model training, we flattened multi-dimensional arrays into one-dimensional formats and normalized these to ensure a consistent scale across all data types, thereby optimizing the performance of subsequent algorithms. This comprehensive approach to data preparation is crucial for accurately predicting and categorizing story points in our models. We

Table 2: Embedded Data

| USER STORY | IMAGE FEATURE | SEVERITY | SP |
|---|---|---|---|
| [-2.21136838e01 9.56352428e02 ...] | [-5.24738908e01 2.34980389e01 ...] | 2 | 2 |
| [-4.05773252e01 1.53721854e01 ...] | - [-5.08334517e-01 2.26934329e-01 ....] | 2 | 1 |
| [-5.83501697e01 4.14541990e01] | - [4.55121100e01 1.26431987e01 ...] | 3 | 2 |
| [-2.50783592e01 1.19310036e01 ...] | - [-5.24738908e01 2.34980389e01...] | 2 | 2 |
| [-2.49652594e01 1.48752362e01 ...] | - [-5.03451347e01 2.49840632e01...] | 3 | 1 |

conducted a correlation analysis to explore hidden relationships among individual parameters, incorporating the calculation of the mean of embeddings into a single numeric metric. We took this approach to reduce the dimensionality of complex data, allowing us to identify patterns more effectively and improve the interpretability of the correlation results. The correlation analysis reveals that the Severity Encoded feature has a strong positive correlation (0.55) with StoryPoint Encoded when included (Figure 2). In contrast, both Story Embedding Mean and Image Feature Embedding Mean exhibit low correlations with StoryPoint Encoded (around 0.06 in Figures 1 and 2), indicating a weaker relationship with the target variable. Despite these differences, XGBoost effectively handles both correlated and non-correlated data (Chen & Guestrin, 2016). Notably, the Story Embedding Mean and Image Feature Embedding Mean are average values representing the embedded features from text data (story descriptions) and image data (visual elements),

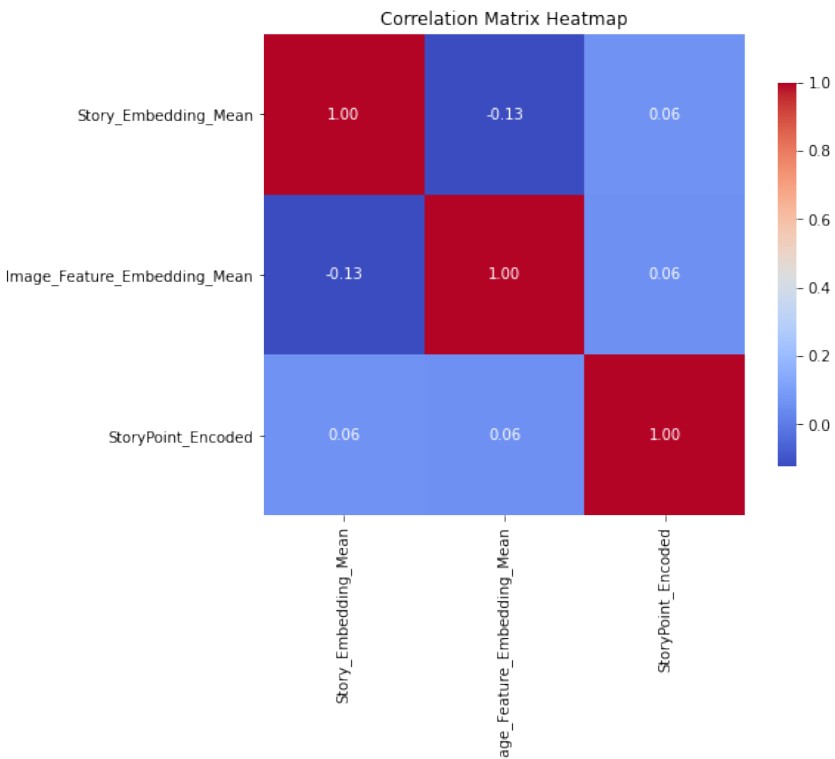

Figure 1: Correlation Analysis with Mean Embeddings Metric - Excluding Severity.

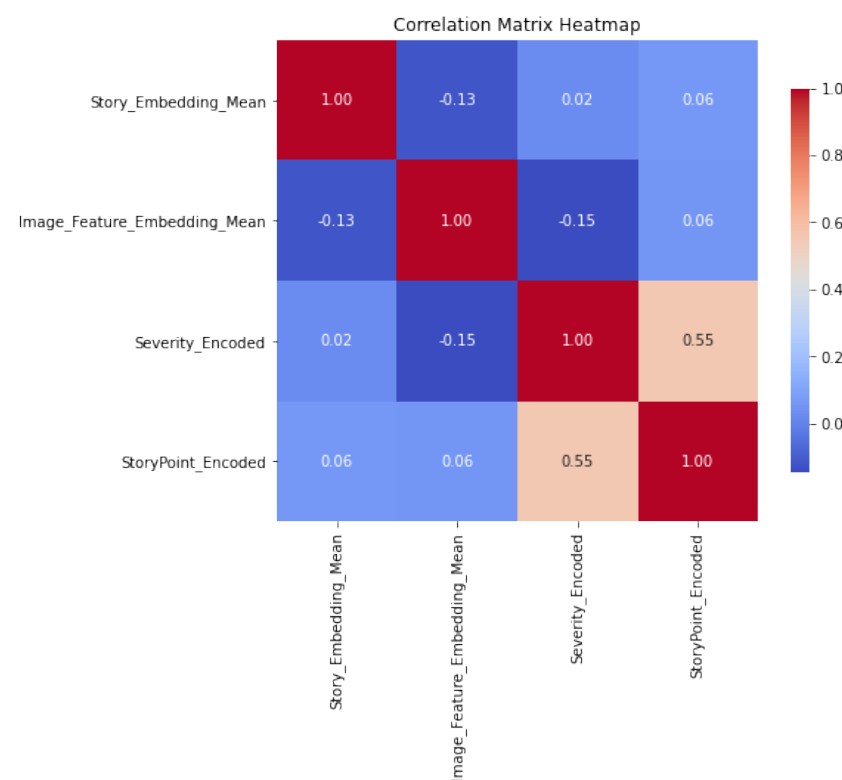

Figure 2: Correlation Analysis with Mean Embeddings Metric – Including Severity

respectively. These means help capture the overall characteristics of the stories and images, aiding in more accurate story point estimation. Without the Severity Encoded feature, the correlations among the other features remain consistent and relatively low, suggesting that these features are largely independent and do not strongly influence the story points on their own. The introduction of Severity Encoded does not significantly alter the relationships between the other features but highlights its importance in the model. Therefore, including Severity Encoded in the model may enhance its predictive accuracy, while the embeddings provide additional, albeit weaker, contributions. However, incorporating severity could also introduce added complexity, which may prevent any noticeable improvements in accuracy.

## 3.3 MODEL DEVELOPMENT AND TRAINING

After integrating BERT text embeddings, CNNextracted image features, and encoded categorical data, we trained a multimodal generative AI model for story point estimation. To assess the significance of severity data in the estimation process, we trained the model both with and without including severity data. The model was designed to learn patterns across the multimodal data—text, images, and categorical values—corresponding to predefined Fibonacci sequence story point classes. We approached the task as a classification problem. We used TensorFlow, a Pythonbased open-source machine learning framework, for all our modeling efforts. For the final estimation of story points, we utilized XGBoost, a powerful ensemble learning algorithm known for its efficiency and performance (Equation 1).

$$\hat{y}_i = \sum_{k=1}^{K} f_k(x_i) \tag{1}$$

Where:

Table 3: User Stories with Severity and Story Points

| PARAMETER | DEFAULT VALUE | FINE-TUNED VALUE | COMMENTS |
|---|---|---|---|
| $n_estimators$ | 100 | 75 | Reduced to prevent over-fitting due to the small dataset. |
| $max_depth$ | 6 | 4 | Lowered to simplify the model and reduce complexity. |
| $learning_rate$ | 0.3 | 0 | Reduced for gradual learning, balancing performance and risk. |
| subsample | 1 | 1 | Introduces randomness to reduce over-fitting. |
| $colsample_bytree$ | 1 | 1 | Helps reduce overfitting by adding feature selection randomness. |
| gamma | 0 | 1 | Increased to make the model more conservative with splits. |
| $min_child_weight$ | 1 | 3 | RIncreased to avoid splits that add little value. |
| $early_stopping_rounds$ | N/A | 15 | Used to prevent overfitting by stopping training early. |

- $\hat{y}_i$ is the predicted value for the $i$-th observation.
- $K$ is the total number of trees (boosting rounds).
- $f_k(x_i)$ is the prediction from the $k$-th tree for the $i$-th observation.

XGBoost was trained on a labeled dataset, with 80 percent of the data used for training and 20 percent reserved for testing to ensure exposure to diverse examples during training. A total of 113 observations were utilized in this process. We adjusted XGBoost parameters for fine-tuning (Table 3).

### 3.4 MODEL EVALUATION AND VALIDATION

After training, we thoroughly evaluated and validated the XGBoost model. We conducted comprehensive verification and validation by comparing the model's predictions with the actual story points assigned by Agile teams. We included evaluation metrics such as precision, recall, F1 score, accuracy, and other relevant measures to ensure a robust assessment of the model's performance.

## 4 RESULTS & DISCUSSION

### 4.1 INTERPRETATION OF RESULTS

When we compare the model's performance with and without severity data, several key trends emerge. The precision, recall, and F1 scores for story point categories 1 and 3 remain consistently high in both models, indicating strong performance in predicting these categories (Figure 3-5). However, excluding severity data leads to a noticeable improvement in overall model accuracy, which increases from 0.63 to 0.77 (Table 4). This improvement is also reflected across the macro and weighted averages, showing more balanced performance across categories.

Story point category 8, which represents more complex or rare story points, shows significant differences. With severity data included, the model fails to effectively predict this category, resulting in a precision, recall, and F1 score of 0.00 (Figure 3). However, excluding severity data, the model's recall for story point category 8 improves to 1.00 (Figure 5), and the F1 score reaches 0.5 (Table 4), though precision remains low at 0.33 (Figure 4). This indicates the model's ability to identify more complex cases, albeit with some inaccuracies. This comparison suggests that while severity data might add complexity, removing it allows the model to generalize better across different categories, particularly improving its performance on rare or complex story points.

Table 4: Comparison of F1 Scores with and without Severity Data

| SP | ACCURACY (With Severity) | F1 (With Severity) | ACCURACY (Without Severity) | F1 (Without Severity) |
|---|---|---|---|---|
| 1 | | 1.00 | | 1.00 |
| 2 | | 0.67 | | 0.71 |
| 3 | 0.63 | 0.63 | 0.77 | 0.84 |
| 5 | | 0.67 | | 0.67 |
| 8 | | 0.00 | | 0.50 |

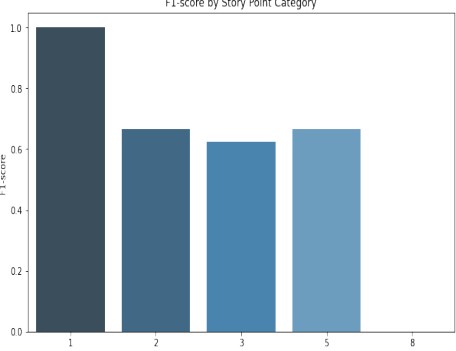 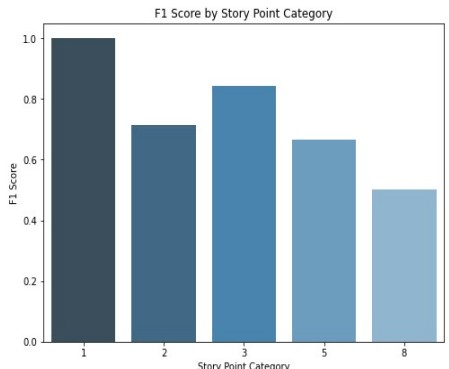

Figure 3: F1 Scores with Severity (left) and without Severity (right)

While the model performed well on simpler categories of story points (1 and 3) in both scenarios, the inclusion of severity data seemed to introduce more complexity than the model could handle effectively, leading to a decrease in overall accuracy and performance balance. The comparison suggests that while severity data may offer additional insights, it also increases the model's complexity, potentially hindering its ability to generalize across all categories.

The confusion matrices further illustrate the model's performance, highlighting that misclassification predominantly occurred in categories with fewer data points, such as category 8. In the first confusion matrix (with severity data), the model shows a tendency to misclassify categories 2 and 3 into one another, but it generally predicts these categories with a reasonable level of accuracy, likely due to the higher number of examples in these categories during training (Figure 6). In contrast, in the second confusion matrix (without severity data), the model displays an improved ability to correctly classify category 3, evidenced by fewer misclassifications, and a better overall performance across categories, especially in handling category 8 (Figure 7). These confusion matrices reflect the challenge the model faces when dealing with imbalanced data, where categories with fewer examples, like category 8, are harder to predict accurately. Additionally, while severity is an influential factor in story point estimation, the improved performance without severity data suggests that other features might be more critical in driving accurate predictions, as severity alone does not account for the complexity of the task. Table 5 compares actual and predicted story points (SP) for 22 user stories, focusing on predictions made with and without considering severity. Notably, certain user stories feature actual and predicted estimations that are very close. In real-life scenarios, development teams often accept estimations as accurate when they fall within a close range. If we applied this approach to the current model, the accuracy would increase to 0.82 when considering severity, and to 0.95 when not considering severity. However, we could still improve the accuracy of these models by training them with a larger dataset, enhancing data preprocessing, and exploring other advanced methodologies.

## 4.2 LIMITATIONS AND CHALLENGES

First, the limited size of the corpus and the imbalance in the dataset, particularly with fewer examples in the higher story point categories, likely contributed to the model's reduced performance in

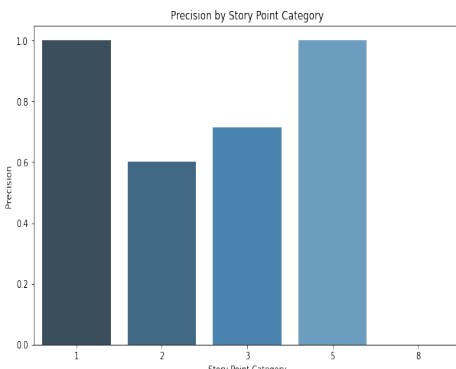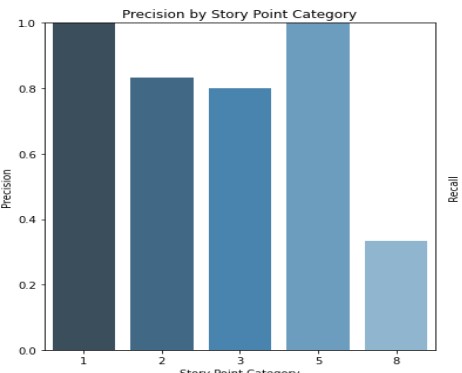

Figure 4: Precision with Severity (left) and without Severity (right)

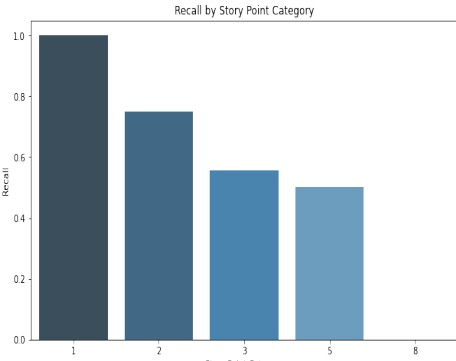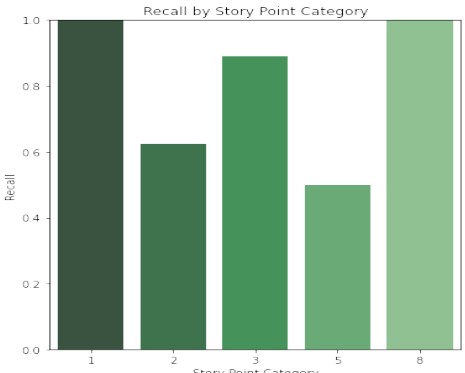

Figure 5: Recall with Severity (left) and without Severity (right)

these areas. This imbalance challenges the model's ability to grasp the nuances of more complex stories, leading to misclassification. Another challenge arises from the integration of multimodal data. Although the combination of text, image, and categorical data provided a more comprehensive feature set, the varying quality and relevance of the image data posed difficulties. Some images, such as architectural diagrams, may not have directly contributed to the estimation process, leading to noise in the data. Moreover, the reliance on BERT embeddings for text representation, while powerful, may have limitations in fully capturing the domain-specific language used in Bugzilla user stories. This limitation could affect the model's ability to generalize beyond the specific dataset used in this study

### 4.3 FUTURE WORK AND IMPROVEMENTS

Future research should address data imbalance by incorporating techniques such as data augmentation or synthetic data generation to provide more examples for underrepresented categories. Additionally, researchers should explore advanced image preprocessing techniques, such as attention mechanisms, to better leverage visual data and reduce the impact of irrelevant images. Another potential improvement involves fine-tuning BERT on domain-specific corpora related to software development and bug tracking. This fine-tuning could enhance the model's understanding of the unique language used in these contexts, potentially improving performance across all story point categories. Additionally, exploring alternative machine learning models or ensemble methods that better handle the complexity and variability of story point estimation could lead to more accurate and reliable results. Integrating these approaches with the current multimodal framework could further enhance the model's robustness and applicability in real-world Agile development settings. Future work should also explore multimodal models such as ViLBERT, CLIP, LXMERT, Visual-BERT, MMT, and others. A larger corpus of pre-processed data is necessary to evaluate how the

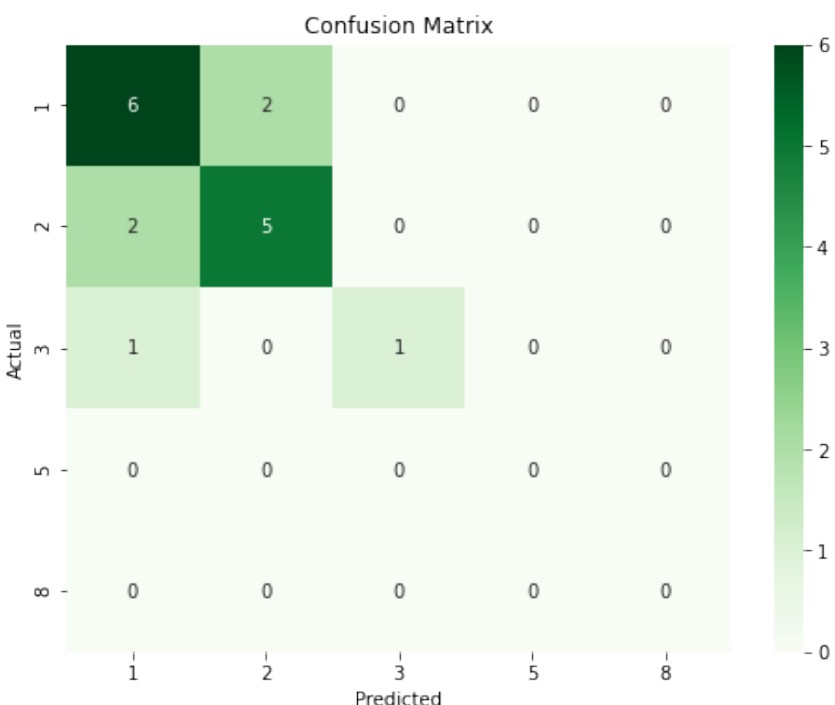

Figure 6: Confusion Matrix with Severity

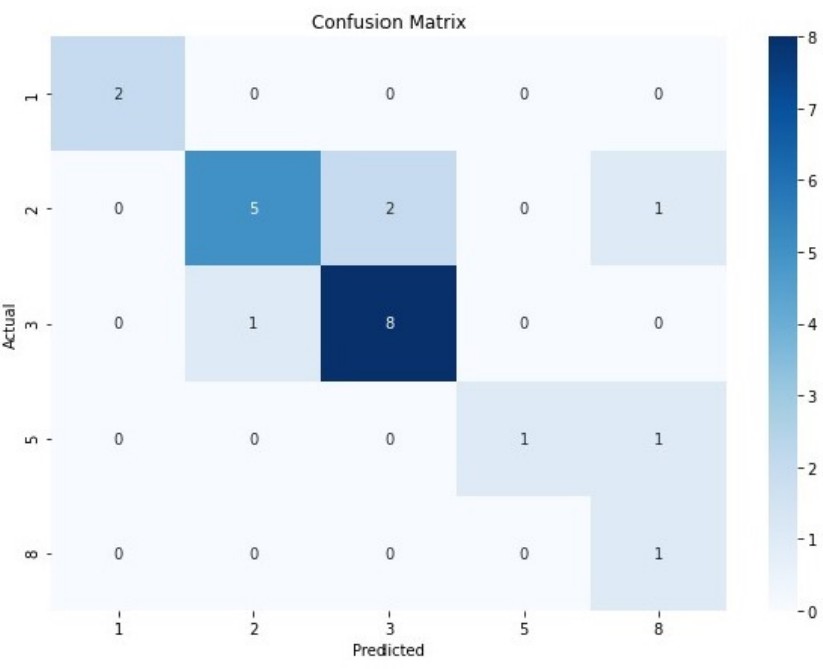

Figure 7: Confusion Matrix without Severity

model performs with a more extensive data pool. Additionally, conducting ablation studies and further analysis on why severity reduced accuracy will be critical for understanding and improving the model's performance.

Table 5: Estimation of User Stories with and without Severity

| USER STORY# | SP | PREDICTED SP WITH SEVERITY | PREDICTED SP WITHOUT SEVERITY |
|---|---|---|---|
| 1 | 3 | 3 | 3 |
| 2 | 2 | 2 | 8 |
| 3 | 2 | 2 | 2 |
| 4 | 3 | 8 | 3 |
| 5 | 3 | 2 | 3 |
| 6 | 2 | 2 | 2 |
| 7 | 3 | 2 | 2 |
| 8 | 3 | 3 | 3 |
| 9 | 5 | 5 | 5 |
| 10 | 1 | 1 | 1 |
| 11 | 2 | 3 | 3 |
| 12 | 2 | 2 | 2 |
| 13 | 2 | 2 | 2 |
| 14 | 3 | 3 | 3 |
| 15 | 3 | 3 | 3 |
| 16 | 2 | 3 | 3 |
| 17 | 8 | 2 | 8 |
| 18 | 3 | 8 | 3 |
| 19 | 1 | 1 | 1 |
| 20 | 3 | 3 | 3 |
| 21 | 5 | 2 | 8 |
| 22 | 2 | 2 | 2 |

## 5 CONCLUSION

This research demonstrated a novel approach to story point estimation in Agile software development by leveraging a Multimodal Generative AI framework. The integration of text, image, and categorical data using advanced machine learning techniques such as BERT for text embeddings, CNN for image processing, and XGBoost for classification has shown potential to improve the accuracy and consistency of story point predictions. The study's main findings highlight that while the model performs well in estimating simpler story points, it faces challenges with more complex categories, particularly due to data imbalance and the varying quality of image inputs.

The significance of this work lies in its contribution to the growing field of AI-driven software project management and development tools. By demonstrating how multimodal data can be effectively integrated to provide more nuanced and accurate estimates, this research opens new avenues for enhancing the efficiency of Agile workflows. The ability to more accurately estimate story points has direct implications for project planning, resource allocation, and overall software development efficiency, making this approach highly relevant to both academic research and industry practices. While this study shows promise, further exploration is needed. Addressing data imbalance, refining multimodal inputs, and tailoring AI models to the language and context of software development are key areas for advancement. Future work should focus on these aspects and extend the approach to other project management domains, bringing us closer to fully realizing AI's potential in transforming Agile software development

ETHICS STATEMENT

This research on Multi-modal Generative AI for Agile software development addresses ethical considerations in AI-driven decision-making, emphasizing the importance of complementing, not replacing, human judgment. Transparency and accountability in AI decisions are key to maintaining trust within Agile teams. Public data from Bugzilla was carefully anonymized to protect privacy. The research also acknowledges potential biases in AI models, particularly regarding data distribution across story point categories, and highlights the need for ongoing efforts to ensure AI tools are fair, transparent, and ethically sound.

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

## A  APPENDIX

### A.1  ALGORITHMS AND MODELS

BIDIRECTIONAL ENCODER REPRESENTATIONS FROM TRANSFORMERS(BERT)

BERT is a transformer-based model that processes text bidirectionally to understand the context of words by considering both preceding and following words. In our research, BERT generates embeddings from user story descriptions, providing rich contextual representations for the story point estimation model. It uses multi-head self-attention mechanisms and is trained on tasks like masked language modeling (MLM) and next sentence prediction (NSP).The equation for the attention mechanism is given below:

$$\text{Attention}(Q, K, V) = \text{softmax}\left(\frac{QK^T}{\sqrt{d_k}}\right)V \tag{2}$$

where $Q$, $K$, and $V$ are the query, key, and value matrices, and $d_k$ is the dimension of the key.

CONVOLUTIONAL NEURAL NETWORK(CNN)

CNNs are deep neural networks used to analyze visual data by extracting spatial features through convolutional layers that detect edges, textures, and other visual elements. In our research, CNNs extract features from images associated with user stories, such as wireframes or screenshots, to enhance story point estimation accuracy alongside text embeddings. The core operation in a CNN is the convolution, defined as (Equation 3):

$$(I * K)(x, y) = \sum_m \sum_n I(x - m, y - n)K(m, n) \tag{3}$$

where $I$ is the input image, and $K$ is the kernel or filter applied to the image to detect features.

XGBOOST (EXTREME GRADIENT BOOSTING)

XGBoost is an optimized distributed gradient boosting library designed to be highly efficient, flexible, and portable. It implements machine learning algorithms under the Gradient Boosting framework, which builds models sequentially, each new model attempting to correct the errors made by the previous ones. XGBoost is used in our research to handle categorical data and produce predictions based on the combined inputs from text and image features.

### A.2  FEATURE ENGINEERING TECHNIQUES

Feature engineering is the process of using domain knowledge to create features that make machine learning algorithms work. In our research, feature engineering involved encoding categorical variables, normalizing text and image features, and integrating them into a cohesive input for the multimodal model. This step is crucial for ensuring that the model can effectively learn from diverse data types.

Techniques Used:

Ordinal Encoding: Used for categorical variables where the categories have a meaningful order.

Normalization: Applied to ensure that features are on a similar scale, particularly when combining data from different modalities.

Embedding Techniques: Used to transform high-dimensional categorical data into lower dimensional continuous vectors.

### A.3  CODE LISTING

The following Python code generates BERT embeddings for tokenized text (Figure 8):

```python
# Step 5: Generate BERT embeddings for the tokenized text in both columns
def get_bert_embeddings(tokenized_text):
    with torch.no_grad():
        outputs = model(**tokenized_text)
        embeddings = torch.mean(outputs.last_hidden_state, dim=1)
    return embeddings

df['story_embedding'] = df['tokenized_story'].apply(get_bert_embeddings)
df['imageFeature_embedding'] = df['tokenized_imageFeature'].apply(get_bert_embeddings)
print(df['story_embedding'].head())
print(df['imageFeature_embedding'].head())
```

```
0    [[tensor(-0.2211), tensor(0.0956), tensor(0.32...
1    [[tensor(-0.0853), tensor(-0.1954), tensor(0.0...
2    [[tensor(-0.4058), tensor(-0.1537), tensor(-0....
3    [[tensor(-0.5835), tensor(-0.4145), tensor(0.3...
4    [[tensor(-0.2508), tensor(-0.1193), tensor(0.0...
Name: story_embedding, dtype: object
0    [[tensor(-0.5247), tensor(0.2350), tensor(0.21...
1    [[tensor(-0.5083), tensor(0.2269), tensor(0.28...
2    [[tensor(-0.5247), tensor(0.2350), tensor(0.21...
3    [[tensor(-0.4551), tensor(0.1264), tensor(0.05...
4    [[tensor(-0.5247), tensor(0.2350), tensor(0.21...
Name: imageFeature_embedding, dtype: object
```

Figure 8: Python code for BERT

The following Python code flattens the numpy arrays of the embeddings, assuming the data is already normalized (Figure 9):

```python
# Convert PyTorch tensor to NumPy array and flatten it
df['story_embedding'] = df['story_embedding'].apply(lambda x: x.numpy().flatten() if torch.is_tensor(x) else x.flatten())
df['imageFeature_embedding'] = df['imageFeature_embedding'].apply(lambda x: x.numpy().flatten() if torch.is_tensor(x) else x.

# Display the first few rows of the processed embeddings
print(df['story_embedding'].head())
print(df['imageFeature_embedding'].head())
```

```
0    [-0.22113684, 0.09563524, 0.3231366, 0.0260801...
1    [-0.08532778, -0.19539368, 0.052737627, 0.1881...
2    [-0.40577325, -0.15372185, -0.29964513, 0.0119...
3    [-0.5835017, -0.414542, 0.33154988, 0.07981408...
4    [-0.2507836, -0.11931004, 0.05581859, 0.122883...
Name: story_embedding, dtype: object
0    [-0.5247389, 0.23498039, 0.21061182, -0.052830...
1    [-0.5083345, 0.22693433, 0.28435063, 0.0658058...
2    [-0.5247389, 0.23498039, 0.21061182, -0.052830...
3    [-0.4551211, 0.12643199, 0.052981365, -0.11214...
4    [-0.5247389, 0.23498039, 0.21061182, -0.052830...
Name: imageFeature_embedding, dtype: object
```

Figure 9: Python code for flattening embeddings

