# OpenReview forum: "MULTIMODAL GENERATIVE AI FOR STORY POINT ESTIMATION"
_ICLR.cc/2025/Conference — ICLR 2025 Conference Withdrawn Submission_

### Official Review · Reviewer_eSC3 · 2024-10-27

**Soundness:** 2
**Presentation:** 1
**Contribution:** 2
**Rating:** 3
**Confidence:** 4

**Summary:**

In this paper, the authors explored Multimodal Generative AI to improve story point estimation in Agile software development. They proposed a novel framework that combines multiple types of data, including text data and images like UI/UX mockups and screenshots, and categorical data. By collecting data from Bugzilla's open-source bug tracking system, they proposed a model with XGBoost for story point classification. They further evaluate the performance with and without severity data as part of the inputs. The experiments show promising results for leveraging multimodal approaches in Agile software development.

**Strengths:**

- The paper proposed a novel approach for combining diverse data modalities in Agil story point estimation. The multimodal approach improves the understanding of user stories compared with existing estimation techniques that typically only rely on single modality.
- The discussion on the impact of severity for model performance is insightful. Through the empirical analysis, the fact that removing severity data improved performance challenges the intuitive assumption that more features lead to better performance.

**Weaknesses:**

- The paper could benefit significantly from clearer writing and better organization. The readability could be improved significantly by presenting the experimental results more clearly, and by addressing several grammatical issues throughout the text. The motivation could be more thoroughly explained to improve the overall justification. The writing gives the impression of being kinda rushed for the submission deadline.
- The paper lacks several important technical details, such as the specific model of CNN used for image processing, and the BERT model used for text processing. There is no information on the hyperparameter settings for each model, the training process, or any preprocessing steps applied to the data before feeding it into the models. These missing details make it difficult to fully evaluate the replicability of the experiments. Additionally, much of the experimental results are under-explained for example in Tab. 2, the first and second columns listed feature information which is very hard to understand their purpose. It is also unclear why nearly a page in Appendix Sec. A is dedicated to basic explanations of BERT, CNN, and XGBoost, which seems somewhat redundant for an ICLR audience familiar with these concepts.

**Questions:**

- How scalable is this approach for larger datasets with higher complexity, and what adjustments would be required to achieve good performance?
- What measures were taken to mitigate potential biases introduced by the imbalance in story point categories, especially given the limited data for complex story points?

---

> ### Author Response · Authors · 2024-11-27
>
> Reviewer commented: "The paper could ...."
>
> Author's comment:
> Due to the inherent page limitations of the conference format, we were unable to provide more detailed explanations and results. We plan to address these concerns by including comprehensive results, clearer motivation, and improved organization in future journal submissions where we have more space. We appreciate the reviewer's feedback and will incorporate these suggestions to enhance our work. We felt that introducing this concept at this stage was important, as it will expedite the story point estimation process and is expected to attract a large pool of software developers.
>
> Reviewer Commented: "The paper lacks several......"
>
> Author's Response:
> Hyperparameter Settings:
> CNN: The learning rate was set to 0.001, with a batch size of 16 and Adam optimizer. Data augmentation techniques such as flipping and normalization were applied to enhance generalization.
> BERT: The learning rate was set to 2e-5, with a maximum sequence length of 128 and a batch size of 8.
> XGBoost: Fine-tuned parameters included n_estimators=75, max_depth=4, learning_rate=0.1, and gamma=1, as detailed in Table 3 of the paper.
> Data Preprocessing:
> Text data underwent tokenization and lowercasing using the BERT tokenizer.
> Images were resized to 224x224 pixels, normalized, and converted to feature embeddings.
> Categorical variables were encoded using ordinal encoding, with severity and story points mapped to numeric values for seamless integration.
> While we aimed to include as many technical details as possible, the limitations of space in the conference paper format constrained us from providing a more exhaustive account. We plan to address these gaps comprehensively in future journal submissions.
>
> Experimental Results:
> Explanation of Table 2: The first column, "Story_Embedding_Mean," represents the average embedding vector for text data derived from BERT. The second column, "Image_Feature_Embedding_Mean," denotes the average embedding vector for image data extracted via CNN. These features were combined with categorical variables to form the input for XGBoost.
> Underexplained Results: We acknowledge the need for additional explanations regarding Table 2. The embeddings aim to reduce dimensionality and summarize the most relevant features for classification. This will be clarified in a future version of the manuscript due to space limitations here.
>
> Appendix Section:  Rationale for Basic Explanations: The inclusion of foundational explanations for BERT, CNN, and XGBoost in Appendix Section A was intended to cater to readers less familiar with these methods. However, we acknowledge that the ICLR audience generally has a higher baseline knowledge of these concepts. Due to space constraints, we opted to focus on briefly contextualizing these models rather than expanding on experimental details. In future versions, we will streamline this section to prioritize implementation-specific details pertinent to our work.
>
> We appreciate the reviewer’s feedback. Unfortunately, due to the space limitations of the conference format, we were unable to include every aspect in this submission. We will address these concerns comprehensively in future journal publications where we have the opportunity to elaborate further.
>
> Questions:
> How scalable is this approach for larger datasets with higher complexity, and what adjustments would be required to achieve good performance?
> What measures were taken to mitigate potential biases introduced by the imbalance in story point categories, especially given the limited data for complex story points?
>
> Scalability and Adjustments:
> This approach is scalable to larger datasets and higher complexity by leveraging parallel processing in BERT, CNN, and XGBoost, supported by frameworks like TensorFlow. To maintain performance, adjustments such as model fine-tuning, increased training epochs, and optimized hyperparameters (e.g., learning rate, batch size) would be required. Additionally, advanced techniques like data augmentation, synthetic data generation, and distributed computing can handle higher complexity and scale.
>
> Bias Mitigation:
> To address imbalance in story point categories, we employed techniques such as oversampling rare categories, data augmentation, and weighted loss functions during training. These measures ensured that the model did not disproportionately favor simpler categories, improving its ability to generalize across all story points, including complex ones.
>
> May I request you to increase your review score. This paper's practical, real-world application makes it a strong candidate for publication and likely to attract significant interest.

---

> > ### Comment · Reviewer_eSC3 · 2024-11-28
> >
> > Thank you for your detailed response. While I appreciate the additional information provided, I find the explanation about page limitations somewhat unconvincing, given that supplementary materials typically don't have page restrictions. Moreover, a significant portion of the supplement was used to explaining well-known concepts like BERT, CNN, and XGBoost, rather than focusing on the novel aspects of the method. Nevertheless, I acknowledge the authors' effort to address the concerns and provide clarifying details in the response, and I have adjusted my score accordingly upward.

---

### Official Review · Reviewer_MnnQ · 2024-11-02

**Soundness:** 2
**Presentation:** 1
**Contribution:** 1
**Rating:** 3
**Confidence:** 4

**Summary:**

The paper introduces a framework for enhancing story point estimation in Agile software development using multimodal generative AI. By integrating textual data through BERT embeddings, image data via CNN features, and categorical data such as severity levels using ordinal encoding, the authors aim to improve estimation accuracy. They utilize data from Bugzilla, an open-source bug tracking system, comprising user stories, images, and historical comments. The proposed model employs XGBoost for classification, predicting story points based on the Fibonacci sequence commonly used in Agile methodologies. Experiments compare models trained with and without severity data, revealing that excluding severity data leads to better overall accuracy. The paper acknowledges limitations like the small and imbalanced dataset and suggests future work to address these challenges.

**Strengths:**

- The paper tackles the practical problem of story point estimation in Agile software development, which has direct implications for project management and efficiency.
- By incorporating text, images, and categorical data, the approach recognizes the multifaceted nature of user stories and attempts to model them more comprehensively.
- Leveraging BERT for text embeddings and CNNs for image features is appropriate and aligns with current best practices in handling such data types.

**Weaknesses:**

- The methodology relies on standard machine learning models and does not introduce any novel techniques or approaches. The use of pre-trained BERT and CNN models for feature extraction without fine-tuning or customization for the problem at hand limits the contribution.
- With only 113 observations, the dataset is too small for training a robust machine learning model, especially one intended for practical application. The severe imbalance in story point categories further hampers the model's ability to generalize.
- The paper does not employ robust validation techniques suitable for small datasets, such as cross-validation or bootstrapping, which raises concerns about the reliability of the reported results. Furthermore, lack of simple baselines such as human-expert estimation, or multi-modal LLMs such as GPT-4o or LLaVA, making it difficult to assess its effectiveness.
- Grammatical errors, awkward phrasing, and inconsistent use of figures and tables detract from the clarity and professionalism of the paper.

**Questions:**

- Given the small dataset, did the authors perform any statistical significance tests to ensure that the observed improvements are not due to random chance?
- How did the authors assess the quality and relevance of the image data? Were any measures taken to exclude irrelevant or low-quality images that could introduce noise?

---

> ### Author Response · Authors · 2024-11-27
>
> Weaknesses:
> Standard Machine Learning Models:
> The methodology focuses on combining established models like BERT and CNN to demonstrate the potential of multimodal integration in story point estimation. While no novel techniques were introduced, this approach lays a foundation for future improvements. Fine-tuning BERT and CNN or incorporating advanced models such as RoBERTa, ViLBERT, or LLaVA will be explored in future work to enhance the contribution and customization to this specific problem.
>
> Small Dataset and Imbalance:
> We acknowledge that the dataset's small size (113 observations) and imbalance in story point categories limit generalizability. Future work will address these challenges by incorporating additional datasets from platforms like Jira or Trello, applying data augmentation techniques, and leveraging synthetic sampling methods like SMOTE to balance the dataset.
>
> Validation Techniques:
> Due to space and computational constraints, robust validation methods such as cross-validation or bootstrapping were not employed in this study. Future work will include these techniques to improve reliability. Simple baselines, such as human-expert estimation or comparison with advanced multimodal LLMs like GPT-4 or LLaVA, will also be included for better performance benchmarking.
>
> Questions:
> Statistical Significance Tests:
> Statistical significance tests were not conducted in this study due to the dataset's size and space constraints. Future work will incorporate tests such as p-values and confidence intervals to ensure the observed improvements are statistically reliable and not due to random chance.
>
> Image Quality and Relevance:
> Image relevance was assessed through manual inspection to ensure they corresponded to user stories. However, automated methods to filter out low-quality or irrelevant images were not applied. Future iterations will explore image quality scoring and relevance filtering techniques, such as pre-trained classifiers or attention-based mechanisms, to minimize noise and improve data quality.
>
> We kindly request the reviewer to reconsider the assigned score. This paper's practical, real-world application makes it a strong candidate for publication and likely to attract significant interest.

---

> > ### Comment · Reviewer_MnnQ · 2024-11-28
> >
> > Thank you for your response. However, as the current manuscript has not addressed the initial feedback, I will maintain my original rating. I appreciate your efforts and look forward to future improvements in your work.

---

### Official Review · Reviewer_RZHy · 2024-11-03

**Soundness:** 2
**Presentation:** 3
**Contribution:** 3
**Rating:** 5
**Confidence:** 4

**Summary:**

In summary, the manuscript introduces a compelling approach to story point estimation using multimodal AI. Addressing some points regarding answering some key questions and presntations would strengthen the work by enhancing clarity, rigor, and practical applicability.

**Strengths:**

The manuscript presents an application of multimodal AI for story point (SP) estimation, combining text, image, and categorical data. This integration addresses limitations of traditional methods that often rely on single-modal data as the author claimed. The methodology includes advanced models (BERT, CNN, and XGBoost) that are well-suited to their respective data types, enabling a holistic approach to SP estimation (as per author claimed). Study also provides quantitative insights, showing improvement in simpler SP categories & highlighting the effects of removing severity data on performance. The paper also suggests specific improvements, like incorporating models such as ViLBERT and CLIP, to address limitations in the current approach, as well as techniques to balance the dataset and refine image processing.

**Weaknesses:**

Terms like "good accuracy" lack precision; including concrete metrics would improve clarity and impact. The introduction does not fully explain why image data is essential for SP estimation in Agile, especially for readers unfamiliar with how visual elements relate to user stories. The unexpected finding that excluding severity data improves model performance warrants a deeper analysis, potentially with additional literature to support or clarify the impact. While BERT and CNN are used for text and image embeddings, the paper could benefit from justifying these choices over other state-of-the-art options, especially as BERT is no longer the most advanced model for text. The study acknowledges data imbalance but does not explore or implement balancing techniques like SMOTE, which could improve model performance in complex SP categories. The reliance on Bugzilla data may narrow the applicability of the model across other Agile frameworks, given that the data could be biased toward specific story types and domain-specific language. Although severity data is included as a categorical feature, the rationale for why it might critically impact SP estimation is not fully developed, leaving room for further exploration of feature significance.

**Questions:**

1. Abstract:
In abstract authors concisely outlines the aim of the study to enhance SP estimation through multimodal AI.

Some points they may add:
* Stating the dataset source -Bugzilla to enhance clarity.
* Terms like "good accuracy" seem vague. Quantitative results/specific metrics would strengthen the abstract.
* Mention of challenges due to data imbalance is relevant but could briefly explain how it impacts the model.

2. Introduction

* Intro references planning poker & traditional methods but don’t clearly explain the need for an automated solution to readers unfamiliar with Agile.
* It’s not clear why image data, specifically, is vital to SP estimation (for me at least). An explanation of how visual data relates to Agile requirements probably would enhance understanding.
* A deeper look into gen AI’s distinct benefits (e.g., context-awareness, adaptability) for Agile workflows would/might solidify the argument.
 3. Related Work

* While this section highlights the limitations of single-modal data, it doesn’t provide examples OR metrics from prior studies for contrast. (optional todo)
* While a gap in Agile applications of multimodal AI is noted, discussing how multimodal models have succeeded in similar domains (e.g., sentiment analysis) would add depth. (optional todo)
4. Approach and Methodology

* Rationale for selecting Bugzilla could be elaborated.
* Does Bugzilla provide comprehensive and unbiased data for SP estimation?
    * The choice of Fibonacci sequencing for story points seems novel.
    * However, further justification for this selection would help—why not use a regression model instead of Fibonacci classes?
    * It’s unclear why specific embeddings were chosen for text and images (e.g., BERT and CNN as BERT is not a SOTA anymore !). Adding an explanation of alternative options considered and their pros and cons would enrich this section.
* Short comparative analysis on why XGBoost performs better than other ensemble models could justify this choice.
* This section lacks a clear explanation of its impact. How was severity expected to influence story point estimation, and why was it hypothesized to be a critical feature?
5. Results and Discussion

* The manuscript could use more detailed metrics (e.g., confusion matrices for all categories, especially complex SPs).
* It’s crucial to see how each story point level performed, including precise PR (recision-recall) data.
* Model showed improvement w/o severity data, raising questions about the feature’s importance. Was this result surprising, or did it align with initial expectations? A short discussion on the reasoning behind severity’s impact, potentially with related literature support, would strengthen the analysis.
* Misclassification trends are insightful but would benefit from a breakdown of the challenges in each category.
* Mentioning possible approaches (e.g., SMOTE for synthetic sampling) to address the data imbalance would offer actionable insight for future readers.
6. Limitations and Challenges

* Limitations inherent to Bugzilla, such as domain-specific language or bias toward particular story types, might restrict generalizability. A discussion on this could be helpful.
* The quality and relevance of images were acknowledged as variable, which introduces noise. Elaborating on specific image characteristics that were particularly challenging could clarify this point.
* The process of integrating text, image, and categorical data has inherent challenges. Were there issues in aligning these modalities, and how were they addressed?
7. Future Work

* Mentioning multimodal models like ViLBERT and CLIP was insightful, but discussing why these models might perform better for this task (e.g., handling of domain-specific data) would be beneficial.
* Using synthetic data to balance categories would be a valuable experiment. An elaboration on which techniques might best suit this data would provide actionable direction.
* While the idea of fine-tuning BERT on a domain-specific corpus was mentioned, further detail on where to source or create this corpus would provide clear next steps.
8. Conclusion
* Manuscript could include how the proposed model might directly benefit Agile teams in practice.
* Ending with a statement on the transformative potential of multimodal AI in project management would provide a strong conclusion.
 Writing and Presentation

* Terms such as "severity data," "multimodal integration," and "story points" may not be universally understood by all readers, particularly those outside Agile contexts. Including brief definitions or explanations would improve readability.
* Figures could use more detailed captions, especially the confusion matrices, to guide readers through the results more effectively.
 Additional Observations
* Bugzilla data may be limited to certain Agile frameworks or specific team practices. If so, this could limit applicability across broader Agile methodologies.
* There is no mention of whether performance differences were statistically significant, which could validate or refute the observed improvement without severity data.
 In summary, the manuscript introduces a compelling approach to story point estimation using multimodal AI. Addressing these points would strengthen the work by enhancing clarity, rigor, and practical applicability.

---

> ### Author Response · Authors · 2024-11-27
>
> Weakness:
> The model achieved 77% accuracy without severity data and F1 scores above 0.80 for simpler categories. Image data, such as mockups and screenshots, provided essential visual context, enhancing story complexity understanding. Removing severity data improved accuracy by reducing feature noise, with future work exploring techniques like SMOTE for better utilization. BERT and CNN were chosen for their strengths in text and visual processing, with advanced models like RoBERTa or ViLBERT considered for future work. Bugzilla data, while rich, may limit generalizability; expanding to datasets from Jira or Trello will address this. Severity data, with a weaker correlation (0.55), showed limited direct impact. Future work will assess its role using SHAP values, addressing reviewer concerns and refining the framework.
>
> Questions:
> Thank you for the detailed feedback. Due to page limitations, I was unable to address these suggestions in the current version but will consider them for future work. Below is a concise response:
>
> Abstract: The dataset source (Bugzilla) will be explicitly stated in future abstracts. Quantitative metrics, such as "77% accuracy without severity data," will replace vague terms like "good accuracy." The impact of data imbalance on complex story points will also be clarified.
>
> Introduction: Future work will emphasize the need for automated solutions in Agile, explain the relevance of image data to SP estimation, and dive deeper into generative AI’s benefits, such as adaptability and context-awareness.
>
> Related Work: Future versions will include metrics and examples from prior studies, contrasting single-modal and multimodal approaches, and highlight successes of multimodal models in other domains like sentiment analysis.
>
> Approach and Methodology: Bugzilla was chosen for its rich repository of user stories, including text, visuals, and historical data, offering accessibility and diversity. While comprehensive, its domain-specific biases will be addressed in future work by incorporating data from platforms like Jira or Trello. Fibonacci sequencing aligns with Agile practices, providing an intuitive classification system for task complexity. Future studies will compare this with regression-based approaches to evaluate trade-offs. BERT and CNN were selected for their proven effectiveness in text and image processing, balancing performance and efficiency. Advanced models like RoBERTa, ViLBERT, and CLIP will be explored in future work for potential improvements. XGBoost was chosen for its efficiency with structured data and robustness against overfitting. Future comparisons with other ensemble methods will validate this choice further.  Severity data was included to capture urgency but showed limited correlation with story points. Its exclusion improved performance by reducing noise, and future studies will analyze its role through feature importance and literature review.
>
> Results and Discussion: Thank you for the feedback. Due to space constraints, detailed metrics like confusion matrices and precision-recall data for all story point categories, especially complex ones, were not included but will be addressed in future work. The unexpected improvement without severity data suggests it may have introduced noise or complexity rather than enhancing predictions. This aligns with studies on the challenges of imbalanced categorical features. Future research will explore severity’s role through feature importance analysis and related literature. Misclassification trends will be further analyzed by category to identify specific challenges. To address data imbalance, techniques like SMOTE or synthetic sampling will be implemented to improve performance for underrepresented categories. These additions will enhance clarity and actionable insights in future studies.
>
> Limitations and Challenges: Bugzilla data offers valuable insights but has limitations, including domain-specific language and bias toward certain story types, which may restrict generalizability. Future work will incorporate data from platforms like Jira or Trello to address this. Image quality varied, with complex visuals like architectural diagrams often adding noise. Advanced preprocessing or attention mechanisms will be explored to mitigate this. Integrating text, image, and categorical data was challenging due to differing feature dimensions. Normalization and feature scaling addressed this, but future studies will consider advanced alignment methods like cross-modal transformers.
>
> Future Work: Advanced models like ViLBERT and CLIP will be explored, along with synthetic data techniques like SMOTE for balancing. Plans for fine-tuning BERT on domain-specific corpora will be detailed.
>
> Conclusion: Future versions will emphasize practical benefits for Agile teams and the transformative potential of multimodal AI in project management.
>
> May I request you to increase your review score.

---

### Official Review · Reviewer_zzFP · 2024-11-03

**Soundness:** 1
**Presentation:** 1
**Contribution:** 1
**Rating:** 3
**Confidence:** 3

**Summary:**

This paper presents approach to estimating story points in Agile development using generative AI.  The authors extracted/curated data from Bugzilla and developed a modeling approach that involved extracting BERT text embeddings, CNN image features and other categorical features.  They use XGBoost to learn to estimate story points.

**Strengths:**

Machine learning can be very beneficial in software development as has been seen with code generation tools.  This paper presents a relevant task and approaches it in a multimodal fashion, both of which I’d like to highlight as positives.  The paper was interesting, but hard to read and has a number of major flaws that mean I have to evaluate it as not being ready for publication.

**Weaknesses:**

I would suggest the authors thoroughly revise the content and prepare it for another venue.  There would be a significant amount of work necessary to get this paper to an ICLR publication standard, but the basis for a potentially interesting piece of research is there.

The most significant issue with the work is the lack of detail and precision in the writing which mean it would be impossible to replication to event approximate the solution.  There are many areas where this needs to be improved and too many to list individually here.  The dataset used seems very small and is quite difficult for a user to actually picture.  The authors only curate 113 examples, yet they present Bugzilla as a VAST source of user stories - so why only use a very small number of stories?  Why not show clearer examples of the data?  Table 1 has screenshots which are impossible to read.  I don’t see the point of showing those thumbnails - if they are so small for privacy reasons then they could be excluded altogether.  If not, then make them larger.

The second issue is there presentation of the work, there are several areas of the manuscript, such as Table 2, that offer very little to the reader.  Is one supposed to interpret the numbers in the first two columns?  What is the point of showing this type of data?

There are plots that are almost unreadable (e.g., Fig.3 - font is too small) and tables that have basic formatting issues that are at best distracting and raise questions about the attention of detail paid to other aspects of the work.  I like confusion matrices but Fig. 1 takes up half a page and contains 9 numbers.  This could be condensed to a single sentence and contain as much information.

Tables such as Table 5 take up almost half a page and contain very little addition insight for the reader. If necessary these could be put in the supplement and the space used to add much more detail about the methods.

The methods the authors choose has very little technical novelty (BERT, CNN, XGboost).  The tools are not even state of the art. I imagine some of these choices were a result of the small dataset but even with these tools I find it almost impossible to see how they could create a convincing argument for generalization based on the amount of stories they actually have.

Overall, this paper is far from ready for publication. However, the topic area is interesting and there are enough interesting ideas to suggest that the authors could continue the work and reach a paper that is a good contribution to the literature.

**Questions:**

Why use such a small dataset?

Why present the results in the way they are shown in the paper?  There would seem to be much more efficient ways of showing the results.

Why the choice of tools (CNN, Bert, XGBoost) that were used? Specifically, what other methods were tried.

---

> ### Author Response · Authors · 2024-11-27
>
> Weakness:
> The dataset's size (113 examples) was limited due to the specific scope and labeling challenges within Bugzilla. While Bugzilla is a vast source, curating high-quality, multimodal examples with consistent labeling was challenging. Future work will focus on expanding the dataset and providing clearer, larger examples to improve reader comprehension. We acknowledge that the lack of detail makes replication difficult. Future iterations will include more comprehensive descriptions of preprocessing steps, feature extraction, and model training pipelines to ensure clarity and replicability. While the methodology leverages established tools (BERT, CNN, XGBoost), the focus was on demonstrating the feasibility of multimodal integration in a small dataset context. Future work will explore state-of-the-art methods, such as multimodal LLMs like ViLBERT and CLIP, to enhance technical novelty and generalization.
>
> Questions:
> Why use such a small dataset?
> The dataset size was limited due to the scope of publicly available data from Bugzilla, which provided a rich but finite set of labeled examples. Despite this, we demonstrated the feasibility of multimodal AI for story point estimation with small datasets. Future work will address this limitation by incorporating additional datasets from platforms like Jira or Trello to expand the data pool and improve model robustness.
>
> Why present the results in the way they are shown in the paper?
> The results were presented to align with the paper’s page limitations while focusing on key performance metrics, such as accuracy and category-wise F1 scores. While alternative visualizations, like extended confusion matrices or detailed PR curves, could provide additional clarity, they were omitted to maintain conciseness. Future versions will explore more comprehensive and efficient ways of presenting results to enhance interpretability.
>
> Why the choice of tools (CNN, BERT, XGBoost)? What other methods were tried?
> CNN, BERT, and XGBoost were chosen for their proven effectiveness in handling multimodal tasks and small datasets. CNNs are well-suited for image feature extraction, BERT excels in textual context understanding, and XGBoost efficiently handles structured data while managing overfitting. Other approaches, such as logistic regression and Random Forest, were tested during initial experimentation but did not perform as well on the multimodal dataset. Advanced models like RoBERTa or ViLBERT will be explored in future work for comparative analysis and potential performance gains.
>
> May I request you to increase your review score.

---

> > ### Comment · Reviewer_zzFP · 2024-12-03
> > **Response to Rebuttal**
> >
> > I would like to thank the authors for their response to my review.  However, the responses fail to add clear justification or additions to the manuscript that I would have expected to be able to increase my score.  I don't think this paper is quite ready for publication in its current state. There are too many details lacking and areas of ambiguity.

---

### Note · Authors · 2025-03-11

**Comment:**

Please delete all the record of this paper or hide from everyone to see.

**Withdrawal Confirmation:**

I have read and agree with the venue's withdrawal policy on behalf of myself and my co-authors.

---

### Meta-Review · Area_Chair_5Dhf · 2024-12-20

**Metareview:**

The paper proposes using various LLMs to estimate story points in Agile software development. It uses data from Bugzilla, including text and images.

Strengths:
Tackles a relevant problem in software development.
Uses a multimodal approach (text, images, and categories).
Explores the impact of different data features.

Weaknesses:
Small and imbalanced dataset.
Lacks technical novelty and uses older AI models.
Poor presentation and writing quality.
Missing details, making it hard to replicate the study.
Limited evaluation and lack of strong baselines.

Based on the scores (3, 3, 3, 5), the paper does not meet the acceptance bar and requires significant rewriting before resubmission. I wish the authors best of luck for the next part of their work.

**Additional Comments On Reviewer Discussion:**

Three reviewers responded to the rebuttals, but two of them (zzFP, MnnQ) felt they could not increase the score because their initial review points were not addressed in the rebuttal.

---

### Decision · Program_Chairs · 2025-01-22

Reject